# Global Transcriptomic Analysis of Topical Sodium Alginate Protection against Peptic Damage in an In Vitro Model of Treatment-Resistant Gastroesophageal Reflux Disease

**DOI:** 10.3390/ijms251910714

**Published:** 2024-10-05

**Authors:** Pelin Ergun, Tina L. Samuels, Angela J. Mathison, Kate Plehhova, Cathal Coyle, Lizzie Horvath, Nikki Johnston

**Affiliations:** 1Department of Otolaryngology and Communication Sciences, Medical College of Wisconsin, Milwaukee, WI 53226, USA; pergun@mcw.edu (P.E.); tsamuels@mcw.edu (T.L.S.); 2Mellowes Center for Genomic Science and Precision Medicine, Medical College of Wisconsin, Milwaukee, WI 53226, USA; amathison@mcw.edu; 3Reckitt Benckiser Healthcare UK Ltd., Slough SL1 3UH, UK; kate.plehhova@reckitt.com (K.P.); cathal.coyle@reckitt.com (C.C.); lizzie.horvath@reckitt.com (L.H.)

**Keywords:** pepsin, gastroesophageal reflux disease, sodium alginate, Gaviscon, reflux, cancer, RNA sequencing, proton pump inhibitors

## Abstract

Breakthrough symptoms are thought to occur in roughly half of all gastroesophageal reflux disease (GERD) patients despite maximal acid suppression (proton pump inhibitor, PPI) therapy. Topical alginates have recently been shown to enhance mucosal defense against acid-pepsin insult during GERD. We aimed to examine potential alginate protection of transcriptomic changes in a cell culture model of PPI-recalcitrant GERD. Immortalized normal-derived human esophageal epithelial cells underwent pretreatment with commercial alginate-based anti-reflux medications (Gaviscon Advance or Gaviscon Double Action), a matched-viscosity placebo control, or pH 7.4 buffer (sham) alone for 1 min, followed by exposure to pH 6.0 + pepsin or buffer alone for 3 min. RNA sequencing was conducted, and Ingenuity Pathway Analysis was performed with a false discovery rate of ≤0.01 and absolute fold-change of ≥1.3. Pepsin-acid exposure disrupted gene expressions associated with epithelial barrier function, chromatin structure, carcinogenesis, and inflammation. Alginate formulations demonstrated protection by mitigating these changes and promoting extracellular matrix repair, downregulating proto-oncogenes, and enhancing tumor suppressor expression. These data suggest molecular mechanisms by which alginates provide topical protection against injury during weakly acidic reflux and support a potential role for alginates in the prevention of GERD-related carcinogenesis.

## 1. Introduction

GERD is a chronic public health issue characterized by symptoms of heartburn and/or regurgitation [1]. GERD is a widespread and common disorder [2]. According to research, it affects ~14% of the global population, with significant variability between regions and countries [3]. In a subset of GERD patients, chronic mucosal exposure to gastric contents prompts a metaplastic transformation of the esophageal lining, culminating in Barrett’s esophagus (BE), a precancerous state [4].

Despite the status of PPIs as the first-line therapy for GERD, 50–55% of GERD patients have symptoms that are resistant to PPI treatment [5,6]. PPIs are strong acid-suppressive drugs, yet they do not offer protection against non-acid substances like bile acids and pepsin, and patients frequently continue to have weakly to non-acidic reflux [6,7,8]. Further, long-term PPI use has been associated with negative effects, including bone fractures, kidney disease, dementia, and gastric neoplasia [9,10]. As a result, patients with GERD have become increasingly likely to decline long-term PPI therapy in lieu of alternative solutions [11,12]. While anti-reflux surgery effectively provides symptom relief [13], candidate selection is limited to patients with pathological GERD (particularly severe erosive esophagitis) [12], and fundoplication failure is not uncommon [6].

Natural seaweed-derived alginates have been shown to inhibit pepsin and sequester bile [14] and have long been used as a therapy for mild and moderate GERD. Preclinical and clinical investigations have established their capacity to prevent reflux-attributed mucosal injury and symptoms [15,16,17,18,19,20]. Alginate-antacid anti-reflux medications create a floating barrier (raft) on top of the stomach contents. This raft eliminates or displaces the post-meal “acid pocket”, thus reducing acidic reflux events [21]. Recent work supports an additional role for commercial alginate-antacid solutions [Gaviscon Advance (GA) and Gaviscon Double Action (GDA)] in topical mucosal protection against injury by refluxate constituents such as pepsin, acid, and bile [15,18,22,23]. Given their capacity for comprehensive protection against acid and nonacid reflux constituents, alginates may be a useful adjunctive therapy for patients experiencing breakthrough symptoms despite maximal acid suppression [24]. While the molecular events leading to mucosal damage during GERD have been the focus of extensive research, esophageal damage caused by weakly acidified pepsin, representative of the reflux of patients with PPI-recalcitrant GERD, is poorly understood. Both the molecular pathways of injury elicited by weakly acidified pepsin in the esophagus and the capacity for alginate to rescue such changes warrant further research.

To better understand the role of sodium alginate in topical mucosal protection against molecular injury during PPI-recalcitrant GERD, we investigated whether alginate could shield against transcriptomic alterations caused by weakly acidified pepsin in normal human esophageal cells.

## 2. Results

Transcriptomic sequencing resulted in a mean total reads per sample of 36,676,200, with a mean of 15,073,461 read gene counts and 77,258,617 read exon counts per sample. Quality control indicated high-quality sequencing data. In principal component analysis (PCA), one sample from GDA+ PA (pepsin-acid) was observed to be significantly different from others and excluded. DE genes were analyzed according to a data-driven approach. Using all detected genes, principal component 1 (PC1) accounted for 65.8% of all gene expression variance, PC2 for 16.7%, and PC3 for 4.5% (Figure 1). The top 100 contributing genes in PC1-3 from PCA analysis of immortalized, normal-derived human esophageal epithelial cells (Het-1A) cells are shown in Appendix A. A heatmap of the conditions and replicates in pairwise comparison are shown in Appendix A. One thousand and twenty-seven differentially expressed (DE) genes were defined as those having a false discovery rate (FDR) of ≤0.01 and an absolute fold-change (FC) ≥ 1.3 (Appendix A).

Complete names of all gene symbols below are provided in Appendix B. Based on Ingenuity Pathway Analysis (IPA) of DE genes, the top canonical pathways included the Keratinization, Collagen Degradation, and Wound Healing Signaling Pathways, and the top disease/disorders included Gastrointestinal Disease, Organismal Injury and Abnormalities, and Cancer (Table 1, Appendix A, *p* ≤ 0.005). The top molecular and cellular functions affected by PA were Cell Morphology, Cellular Development, Gene Expression, Cellular Growth and Proliferation, and Cellular Assembly Organization (Table 1 and Appendix A). Top upstream regulators *POU2F3*, *SPRR5*, *EFNA3*, and *PAX1* were common across the comparisons of Sham + PA vs. Sham + Sham, Placebo + PA vs. Sham + Sham, and Placebo + PA vs. Sham + PA (Table 1). In the comparison of Gaviscon formulations (GA and GDA) + PA to Sham + PA and Placebo + PA treatments, top upstream regulators included those involved in posttranslational modification *SMYD3*, *HIPK1*, *SPOP*, *BRAF*, *Firre*, and *miR-3648*; signaling molecules *TLR5*, *IL2RG*, *IL4*, and *BRAF*; and transcription factors *HNRNPK* and *POU2F3* (Table 1).

IPA Tox Function analysis identified biological mechanisms related to toxicity at molecular, cellular, and biochemical levels. Tox Function Pathways were downregulated in these comparisons, as evidenced by the -log (*p*-value) ≥ 1.3. The analysis revealed that Transforming Growth Factor Beta (TGF-β) Signaling Pathway inhibition was observed in cells treated with PA following GA or GDA pretreatment compared to both Sham + PA and Placebo + PA groups (Figure 2 and Appendix A) (FC ≥ 1.3). Mechanisms involving Gene Regulation by Peroxisome Proliferation via the Peroxisome proliferator-activated receptor α (PPARα) Pathway were decreased in GA + PA vs. Sham + PA, GDA + PA vs. Sham + PA, and GA + PA vs. Placebo + PA (Figure 2 and Appendix A).

Ranked by FDR, the top ten up- and downregulated DE genes for each comparison are listed in Table 2 and Appendix A. *MT-ND6*, *FAT4*, and *LPP* were commonly upregulated genes in both GA + PA and GDA + PA groups relative to Sham + PA or Placebo + PA. *MZT2A*, *PCSK1N*, *FBXL15*, and *SCAND1* were commonly downregulated genes in both Gaviscon treatments relative to Sham + PA and Placebo + PA. Proto-oncogenic *JUN* was significantly decreased in GDA + PA versus both Placebo + PA and Sham + PA groups. Placebo + PA decreased *FAT4*, *FAT1*, and *LPP* genes relative to Sham + Sham treatment and decreased the cilia structure gene *DNAAF4* and ion channel activity gene *TRPC3* relative to Sham + PA. Relative to Sham + Sham, PA treatment upregulated *SMOX* and *AGER* and downregulated *KRT14*, *KRT13*, and *FAT4* genes.

## 3. Discussion

In untreated GERD patients, refluxate is predominantly acidic (63% pH < 4); however, a significant percentage of reflux events are weakly to non-acidic [25,26]. While PPIs reduce the acidity of refluxate, they do not reduce the frequency of reflux events [27]. Continued weakly or non-acidic reflux is thought to contribute to persistent symptoms exhibited by a large proportion of GERD patients despite PPI therapy [5,6,7,26,28]. Nonacid reflux constituents are also suspected to underly reflux-attributed esophageal carcinogenesis, as indicated by the rising incidence of esophageal adenocarcinoma despite widespread use of PPIs and the association between long-term high adherence to a PPI regimen and greater risk of carcinogenesis and high-grade dysplasia [29]. All gastric juice and refluxate contains pepsin, irrespective of PPI use [8,30]. Evidence demonstrating pepsin’s role in reflux-attributed diseases has developed significantly in recent decades, particularly where it relates to the airways [8,31,32,33]. It is well known that although pepsin’s enzymatic activity decreases as pH levels rise, the enzyme remains stable up to a pH of 8 [34]. While pepsin is enzymatically inactive at neutral pH, it is taken up by airway and esophageal epithelial cells through receptor-mediated endocytosis and is stored in intracellular vesicles of pH 4–5 where it would be reactivated [35,36]. We have previously demonstrated molecular mechanisms of injury by nonacid pepsin in esophageal cells that are consistent with inflammatory and carcinogenic processes attributed to GERD [31,37,38]. The erosive and carcinogenic effects of pepsin can be attenuated by pepsin-inhibiting compounds [15,18,32,39]. Most in vitro research investigating molecular injury during GERD to date has focused on acid alone, or in some cases acidified pepsin [30,40]. Little to no research has been conducted to investigate potential esophageal injury by weakly acidified pepsin. Scientific inquiry to investigate mucosal injury by refluxate pH > 4 is critical to improve our understanding of PPI-recalcitrant GERD.

Clinical studies have shown that combined alginate-antacid formulations can relieve GERD symptoms and provide mucosal protection [41,42,43]. The antacid in these formulations neutralizes acid while the alginate inhibits the enzymatic activity of pepsin and sequesters bile salts [16]. Recent work indicates that alginate formulations provide topical mucosal protection [15,18,22,23]. However, preclinical studies are necessary to better understand how topical alginate may protect against direct and indirect injury to the esophagus caused by gastric contents [44,45,46,47]. Here, we have utilized RNA sequencing to identify molecular pathways altered by pepsin and weak acid in a cell culture model of PPI-recalcitrant GERD and their potential rescue by GA and GDA pretreatment.

### 3.1. Genes Differentially Expressed by PA Treatment

Our findings indicate that pepsin combined with weak acid caused collagen degradation and keratinization, as indicated by prominent canonical pathways in IPA. *KRT13*, *KRT14*, small proline-rich proteins (SPRRs), *COL17A1*, and *MMP-10* were decreased in PA-treated cells relative to sham (Table 2) in accordance with similar findings in the current literature [48,49]. Previous studies have shown that pepsin and weak acid causes depletion of transcripts that encode structural or extracellular matrix components [31,50,51].

In accord with previous in vitro findings [31,32,38,52], PA treatment yielded significant changes in genes with roles in chronic inflammation and cancer development (Table 2). *SMOX* oxidizes spermine to spermidine and generates reactive oxygen species that cause oxidative-stress-induced apoptosis, increase DNA damage, and promote tumorigenesis [53,54,55,56,57,58]. A non-specific multi-ligand receptor, *AGER (RAGE)*, is involved in cellular cascade responses leading to inflammation both in vitro and in vivo and is implicated in the pathophysiology of various diseases [59,60]. Along with its pro-inflammatory role, *AGER* overexpression promotes migration, invasion, and epithelial–mesenchymal transition (EMT), features of cancer cells, through ERK signaling [61]. Studies indicate that *AGER* is overexpressed in various types of malignant tumors, including esophageal and gastric cancers [62,63,64]. Meanwhile, the downregulation of notable tumor suppressors such as *FAT4* [65] and *CRNN* [66] supports the cancerous characteristics of pepsin. The loss or mutations of both genes is related to tumorigenesis in esophageal cells [67,68,69].

PA-induced DE genes also included numerous non-protein coding transcripts that are poorly understood or completely uncharacterized. Mining these findings could yield important insights into novel mechanisms of peptic injury during GERD.

### 3.2. Alginate Rescue of PA-Induced DE Genes Relative to Placebo or Sham

Collagen degradation was the top canonical pathway attributed to PA treatment; dysregulation of this pathway was rescued by GA but not GDA or placebo (Table 1). Collagen degradation and collagen synthesis (including trimerization) are parts of the extracellular matrix (ECM) remodeling process involved in repair and rebuilding of the ECM after reflux-attributed injury [70,71,72].

Tox function analysis revealed that alginate formulations similarly inhibited injury-associated pathways, such as TGF-β signaling and peroxisome proliferation, via PPARα (Figure 2 and Appendix A). TGF-β is a key regulatory cytokine involved in numerous cellular processes, including chronic inflammation, proliferation, differentiation, and ECM remodeling [73,74]. Its altered activity contributes to the development of tissue remodeling and progression to malignancy, such as during development of gastrointestinal cancers [74,75], leading to clinical trials investigating inhibition of TGF-β signaling [75,76,77]. Interestingly, *TGF-β* is overexpressed in the metaplastic stages of esophageal cancer but not in non-neoplastic precancerous cells, such as Barrett’s esophagus (BE) [74,78,79]. However, prolonged inflammatory response may induce the carcinogenic effect of canonical TGF-β signaling in BE or ulcerative colitis [80]. The inhibition of this canonical pathway by alginate relative to sham or non-alginate placebo suggests that alginate may impede the carcinogenic potential of pepsin [31,32,38,52,81,82].

Peroxisome proliferation via the PPARα pathway was the most prominently affected cellular pathway according to IPA Tox Function analysis in GA + PA vs. Sham + PA, GDA + PA vs. Sham + PA, and GA + PA vs. Placebo + PA comparisons (Figure 2 and Appendix A). The pathway plays several common roles in lipid metabolism, energy homeostasis, inflammation regulation, and cellular detoxification [83] and is involved in different types of cancer progression in complex ways [84,85]. PPAR-mediated transcription occurs when a member of the RXR family interacts with a peroxisome proliferator response element within target genes, triggering transcriptional activation or repression [86]. The Liver X Receptor-Retinoid X Receptor (LXR/RXR) activation pathway was significantly decreased in the GA + PA comparison relative to the Placebo + PA comparison. This pathway plays a role in the regulation of lipid metabolism, cell energy, inflammation, cell detoxification, and various types of cancer [87,88,89].

Our study also showed that GDA pretreatment inhibited the Liver Steatosis Pathway compared to the placebo (GDA + PA vs. Placebo + PA) (Appendix A). Steatotic liver disease involves having excess fat in the liver and causes chronic inflammation [90]. A recent fatty liver rat model study showed that sodium alginates recover the liver by modulating gut microbiota and suppressing inflammation via the TLR4/NF-κB/NLRP3 inflammatory pathway [91]. Another mice study revealed that alginates prevent liver inflammation and lipid accumulation in the liver. Furthermore, the upregulation of zonula occludens-1 expression showed that intestinal barrier function was impaired despite the low absorption of alginates by the gastrointestinal tract [92]. These pathway analyses demonstrate the inefficacy of the placebo in providing anti-inflammatory and anti-cancer support, highlighting the protective effects of alginate rather than the solution’s viscosity, as supported by the literature [15,18,23,93]. GDA pretreatment also decreased the Mitochondrial Dysfunction Pathway compared to the Sham + PA group (Figure 2). The results support sodium alginate’s homeostatic effects on cell viability [15].

Alginate formulations upregulated tumor suppressors, cell adhesion, signal transduction, regulatory genes, and non-protein coding genes (Table 2, Appendix A). *FAT4* (GA + PA vs. Sham + PA, GDA + PA vs. Sham + PA, and GDA + PA vs. Placebo + PA) and *FAT1* (GA + PA vs. Sham + PA and GA + PA vs. Placebo + PA) were the top increased tumor suppressors based on FDR. Both are members of the cadherin superfamily and function as adhesion molecules and signaling receptors [94,95]. Many studies have shown the tumor-suppressing role of *FAT1* in esophageal cancer cell lines and esophageal squamous cell carcinoma (ESCC) tissues [94,96,97,98]. Furthermore, lower levels of circular *FAT1* have been suggested as a novel prognostic marker in ESCC patients [99]. *FAT4* is suppressed in various cancer types such as esophageal, gastric, breast, colorectal, liver, and adrenocortical cancer [65,95,100,101]. These two genes were significantly decreased by the direct effect of pepsin and acid treatment compared to untreated cells (Table 2). Pepsin-acid treatment following Gaviscon pretreatments also indicated upregulation of other tumor suppressors, including *BRCA2*, *KLF10*, *SPRY4*, *HERC1* (also increased in GDA + PA vs. Sham + PA), *SHANK2*, *RP11*, and *AHNAK*, relative to Placebo + PA or Sham + PA (Table 2, Appendix A).

Alginate formulations downregulated two important proto-oncogenic genes, *JUN* (GDA + PA vs. Placebo + PA or Sham + PA) and *JUNB* (GA + PA vs. Placebo + PA) (Table 2, Appendix A). Proto-oncogenes are crucial genes involved in cell growth, differentiation, and survival. They encode proteins that regulate cell division and signal transduction pathways. Several different oncogenes are derived from proto-oncogenes [102]. Overexpression of *JUN* is detected in patients with laryngeal squamous cell carcinoma and could be related to progression due to its overactivation with *c-Fos* [103]. Additionally, *JUN* has inflammatory roles when epithelial cells are exposed to gastric fluids. ERK/c-Jun signaling has been found to regulate the expression of *MMP-7* and the degradation of *E-cadherin* in human pharyngeal epithelial cells [104]. *JUNB*, a member of the *JUN* family, is associated with poor prognosis in human gastric cancer and prostate cancer cell lines [105,106]. Beyond its role in cancer, overexpression of *JUNB* may contribute to various inflammatory diseases [107,108].

Although there is limited research, the loss of *MZT2A* expression in alginate formulations relative to both placebo and sham PA groups suggests the importance of this gene in rescue of peptic injury. *MZT2A* controls cell septation signaling during cell development by balancing microtubule polymerization and depolymerization at the centrosome [109]. It is primarily studied in non–small-cell lung cancer, where it is highly expressed in cancer cells compared to normal bronchial cells [109]. Its paralog *MZT2B* (~96% identity) [110], is overexpressed in patients with gastric cancer [111]. Gaviscon formulations also revealed the downregulation of various genes such as *COL6A2* [112], *DOHH* [113], *MEX3D* [114], *SBNO2* [115], which have roles in cancer or chronic disease progression (Table 2, Appendix A). Further research is needed to better understand the potential role of these mechanisms in reflux-attributed esophageal injury.

Wound healing is a complex, well-coordinated process that includes three phases: inflammation, proliferation, and remodeling. To ensure proper healing following an injury, these processes must be strictly managed. Any dysregulation can lead to incomplete healing, chronic inflammation, excessive tissue damage, and even malignancy [116,117]. While *PTGS2* (also known as cyclooxygenase-2, *COX-2*) is primarily known for its pro-inflammatory roles, it is also a well-known mucosal repair gene [118,119]. Studies have revealed that *COX-2* signaling is essential for gastrointestinal epithelial healing. Inhibition of *COX-2* can delay gastric ulcer healing and cause significant damage [120,121]. In our study, pepsin and pH 6.0 acid treatment following GA elevated *PTGS2* expression relative to Placebo + PA treatment (Appendix A).

Similar to the results of Tox function and Canonical pathway analysis, the top downregulated genes were consistent between comparisons of GDA + PA with both Sham + PA and Placebo + PA (Table 1 and Table 2, Figure 2, Appendix A). Further, the Placebo + PA versus Sham + PA comparison demonstrated elevation of certain oncogenes *(MMP1* [122], *NOTCH2NL* [123], *LIPH* [124], and *HBQ1* [125]). These results demonstrate the inefficacy of placebo relative to alginate.

### 3.3. Limitations and Summary

In summary, our study investigates the molecular impact of pepsin and weak acid on normal esophageal epithelial cells and evaluates the protective effects of Gaviscon formulations (GA and GDA) against pepsin-induced dysregulation in GERD-related signaling processes. Using RNA sequencing, we identified significant gene expression changes and pathways associated with inflammation, apoptosis, ECM remodeling, and carcinogenesis. Pepsin exposure notably increased the expression of oncogenic or pro-inflammatory genes while downregulating key tumor suppressors. These data support pepsin’s direct deleterious effect on epithelial barrier integrity and chromatin stability and promotion of a cancer-associated phenotype [15,30,38,126]. Gaviscon formulations demonstrated protective effects by mitigating these detrimental changes, promoting ECM repair, downregulating proto-oncogenes, and enhancing tumor suppressor gene expression. Gaviscon formulations abrogated peptic stimulation of these cancer-associated pathways. These findings provide credence to existing clinical and preclinical evidence that topical sodium alginate protects the esophagus. GDA pretreatment was predominantly associated with RNA regulation and cell signaling, whereas rescue by GA was associated with cell barrier, cell junctional, and lipid metabolism processes. One could surmise that differences between the benefit offered by the different formulations may be related to differences in sodium alginate content; GA has greater sodium alginate content than GDA. While the two formulations demonstrated differences in their ability to rescue certain PA-affected pathways and gene expressions, both formulations conferred benefit relative to placebo, as indicated by comparisons between Gaviscon treatment groups and placebo or sham groups, which demonstrated minimal differences in Tox function analysis, top canonical pathways, top upstream regulators, and significant genes. Our dataset additionally revealed substantial dysregulation of long noncoding RNAs and miRNAs that were dysregulated by PA and rescued by alginate, whose functions in peptic injury of the esophagus warrant further investigation.

There are some limitations to this study. Our dataset includes a single cell culture line (Het-1A) and pH (6.0). Different epithelial cell lines may yield different results and may not adequately represent the clinical situation. Additionally, this model does not account for the potential effects of *Helicobacter pylori* infection, which is known to exacerbate GERD symptoms and may influence mucosal integrity and inflammation. There is emerging evidence that sodium alginate could have a beneficial effect on *H. pylori*, a bacterium associated with gastritis and peptic ulcer disease, potentially reducing its impact on GERD progression. The exclusion of *H. pylori* from the model limits our understanding of how alginate formulations might interact with both GERD and *H. pylori*-related pathology. Moreover, the study reflects only a descriptive analysis, and no functional validation of the transcriptomic outcomes has been performed. Despite these limitations, the molecular mechanisms identified in this in vitro model may have important implications for GERD patients. Pepsin’s ability to dysregulate oncogenic, inflammatory, and ECM-related pathways suggests a potential role in driving disease progression in acid-suppressed patients with persistent weak acid reflux. The protective effects of Gaviscon formulations observed in vitro may translate into clinically relevant benefits by reducing epithelial damage and possibly lowering the risk of progression to more severe conditions, such as Barrett’s esophagus or esophageal adenocarcinoma. Further clinical studies are warranted to validate these protective effects in human populations. The work herein should be interpreted with this in mind, and results warrant corroboration in vivo and in clinical specimens. Despite these limitations, these data provide preclinical evidence that Gaviscon formulations may provide protection against esophageal injury and carcinogenesis that may be promoted by persistent mucosal exposure to weakly acidified pepsin in acid-suppressed GERD patients.

## 4. Materials and Methods

### 4.1. Cell Culture

Immortalized, normal-derived human esophageal epithelial cells (Het-1A) (American Tissue Culture Center, Manassas, VA, USA) were cultured in BEGM (Bronchial Epithelial Cell Growth Medium) medium (Sigma Aldrich, St. Louis, MO, USA). The medium contained the provided supplements of hydrocortisone, insulin, and transferrin, along with 180 μM adenine, 10 ng/mL cholera toxin (both from Sigma-Aldrich), 70 μg/mL bovine pituitary extract, 5% fetal bovine serum, and 1x Antibiotic-Antimycotic (ThermoFisher Scientific, Waltham, MA, USA). The cells were grown on collagen-I coated flasks (Biocoat; Corning, Corning, NY, USA). Once the cells reached 75% confluency, they were subjected to pretreatment conditions in triplicate wells.

The test substances included Gaviscon Advance^®^ (Reckitt Benckiser Group, Slough, UK; containing 1000 mg sodium alginate and 200 mg potassium hydrogen carbonate per 10 mL), labeled as “GA”, and Gaviscon Double Action^®^ (Reckitt Benckiser Group; containing 500 mg sodium alginate, 213 mg sodium bicarbonate, and 325 mg calcium carbonate per 10 mL), labeled as “GDA”. The placebo was a xanthan gum solution with matching viscosity but without alginate or bicarbonates [23]. Pepsin and pH 6.0 treatments were labeled “PA”.

The cultures were pretreated for 1 min in HBSS (Sham) (pH7.4; control, ThermoFisher Scientific, Waltham, MA, USA) or 1:20 dilutions of GA, GDA, and the viscosity-matched placebo control in HBSS. Following pretreatment, the cultures were washed twice with HBSS to mimic the physiologic conditions of esophageal clearance and treated in HBSS pH 7.4 or HBSS pH 6.0 ± 0.1 mg/mL porcine pepsin (Sigma-Aldrich, St. Louis, MO, USA) at 37 °C/5% CO_2_ for 3 min. After treatment, the cells were rinsed twice with HBSS and allowed to recover in normal growth media under the same conditions for 1 h before harvesting for total RNA extraction. Total RNA was extracted using the RNEasy Plus Mini Kit (Qiagen, Hilden, Germany) which eliminates genomic DNA and QIAshredder columns (Qiagen, Hilden, Germany). RNA quality was assessed by UV spectroscopy (Nanodrop 2000; ThermoFisher, Scientific, Waltham, MA, USA), fluorimetry (Qubit; Thermo Fisher Scientific, Scientific, Waltham, MA, USA), and a fragment analyzer (Agilent, Santa Clara, CA, USA) using high-sensitivity RNA components (Figure 3).

### 4.2. RNA Sequencing and Pathway Analysis

RNA libraries were prepared (Illumina TruSeq Stranded mRNA, dual indexed, (Illumina, San Diego, CA, USA) and sequenced on the Illumina Nova Seq 6000 (Illumina, San Diego, CA, USA) with 100 bp paired end reads at the Genomic Sciences and Precision Medicine Center (RRID:SCR_022926), as previously described [126]. Briefly, quality control checks were performed using FastQC (v0.11.9, Babraham Bioinformatics, Babraham Institute, Cambridge, UK) and RSeQC (v3.0.0, developed by Dr. Liguo Wang, Mayo Clinic—Rochester, NY, USA). The MAPRSeq3 suite (http://bioinformaticstools.mayo.edu/research/maprseq/, accessed on 1 October 2024) was utilized for analyzing paired-end RNA-Seq data: read alignment was conducted with Star (v2.5.2b, developed by Dr. Alexander Dobin, Cold Spring Harbor Laboratory, Cold Spring Harbor, NY, USA), BAM files were processed using featureCounts (v1.5.1, Walter and Eliza Hall Institute of Medical Research, Parkville, Australia, and exon quantification was obtained via BEDTools (v2, developed by Quinlan Lab, University of Utah, Salt Lake City, UT, USA). This process resulted in gene and exon counts, both raw and normalized for sequencing depth and gene length (linear reads per kilobase of transcript per million). Differential expression (DE) analysis was executed using a pairwise approach with EdgeR (developed by the Walter and Eliza Hall Institute of Medical Research, Parkville, Australia), employing the following thresholds: a minimum of one read per million in at least three samples, a false discovery rate (FDR) of ≤0.01, and an absolute FC ≥ 1.3 [log2(FC) ≥ 0.3785]. Further analysis of differentially expressed genes was conducted using Ingenuity Pathway Analysis (IPA; Qiagen, Hilden, Germany) and principal component analysis (PCA).

## 5. Conclusions

Our study reveals the molecular impact of pepsin and weak acid on normal esophageal epithelial cells and evaluates the protective effects of sodium alginates against pepsin-induced dysregulation in GERD-related signaling pathways. RNA sequencing identified significant gene expression changes associated with inflammation, apoptosis, ECM remodeling, and carcinogenesis. Pepsin exposure notably increased oncogenic and pro-inflammatory genes while downregulating tumor suppressors, highlighting its detrimental effects on epithelial barrier integrity and chromatin stability. Gaviscon formulations mitigated these harmful changes, promoting ECM repair, downregulating proto-oncogenes, and enhancing tumor suppressor gene expression. The protective effects were attributed to sodium alginate’s mucoadhesive, pepsin-inhibiting, and acid-neutralizing properties, as evidenced by the relative inefficacy of viscosity-matched placebo. Our findings support the role of sodium alginate in protecting the esophagus during PPI-recalcitrant GERD and emphasize the need for further research to explore new therapeutic approaches for this patient population.

## Figures and Tables

**Figure 1 ijms-25-10714-f001:**
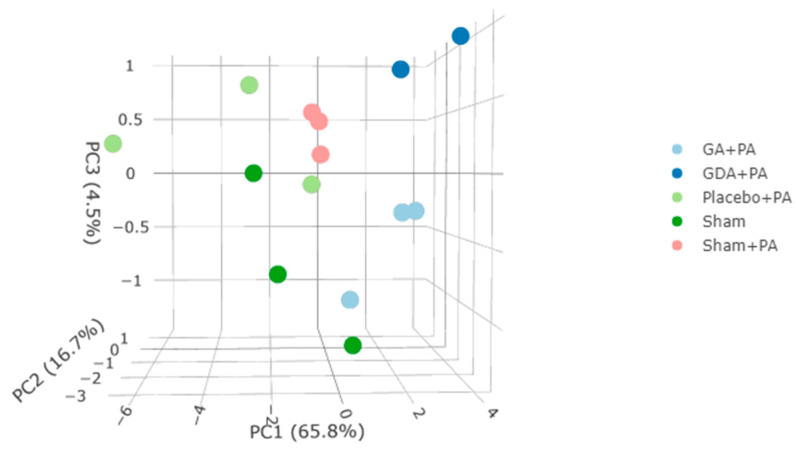
PCA of RNA sequence data of Het-1A cells treated with Gaviscon Advance + PA (light blue), Gaviscon Double Action (dark blue), a matched placebo control + PA (light green), pH 7.0 buffer (sham) (dark green), and sham + PA (pink). PC1 accounted for 65.8% of the variance, PC2 for 16.7%, and PC3 for 4.5%. PCA = principal component analysis; GA = Gaviscon Advance; GDA = Gaviscon Double Action; PA = Pepsin + acid, Het-1A = immortalized, normal-derived human esophageal epithelial cells.

**Figure 2 ijms-25-10714-f002:**
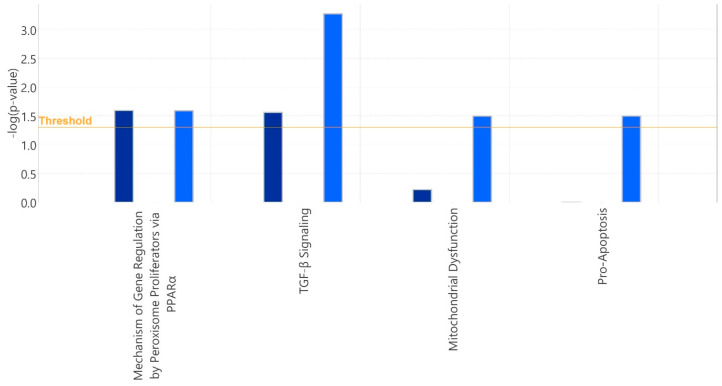
IPA Tox Function analysis. Dark blue: GA + PA vs. Sham + PA. Blue: GDA + PA vs. Sham + PA. Blue color indicates z-score is negative. GA = Gaviscon Advance; GDA = Gaviscon Double Action; PA = pepsin + acid.

**Figure 3 ijms-25-10714-f003:**
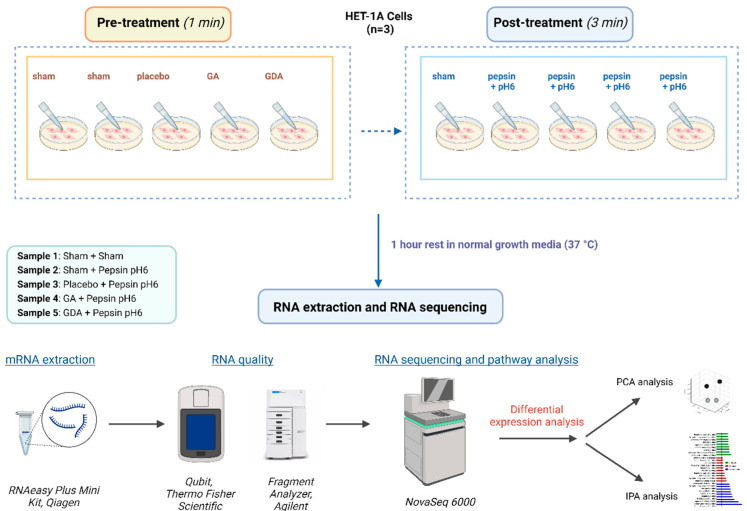
Research workflow. Sham: pH7.4 HBSS, Placebo: xanthan gum solution with matching viscosity but without alginate or bicarbonates, GA = Gaviscon Advance; GDA = Gaviscon Double Action; PA = pepsin + acid.

**Table 1 ijms-25-10714-t001:** Ingenuity Pathway Analysis of significant DE transcripts.

**Sham + PA vs. Sham + Sham**		
**Top Canonical Pathways**	***p*-Value**	**Overlap**
Keratinization	7.01 × 10^−7^	2.8% 6/214
Collagen Degradation	3.74 × 10^−3^	3.3% 2/61
Melatonin Degradation II	5.95 × 10^−3^	25.0% 1/4
**Top Diseases**	***p*-Value Range**	**Number of Molecules**
Gastrointestinal Disease	4.78 × 10^−2^–2.28 × 10^−8^	33
Organismal Injury and Abnormalities	4.95 × 10^−2^–2.28 × 10^−8^	36
Cancer	4.73 × 10^−2^–4.19 × 10^−5^	36
**Molecular and Cellular Functions**	***p*-Value Range**	**#Number of Molecules**
Cell Morphology	4.95 × 10^−2^–2.67 × 10^−5^	11
Cellular Development	4.80 × 10^−2^–1.46 × 10^−4^	21
Cellular Growth and Proliferation	4.80 × 10^−2^–1.46 × 10^−4^	20
Cell Death and Survival	4.89 × 10^−2^–4.08 × 10^−4^	18
Cellular Assembly and Organization	4.66 × 10^−2^–4.45 × 10^−4^	14
**Top Networks**		**Score**
Gastrointestinal Disease, Organismal Injury and Abnormalities, Cardiovascular System, Development and Function		56
Cancer, Organismal Injury and Abnormalities, Cell Death and Survival		24
**Top Upstream Regulators**	***p*-Value Range**	
** *POU2F3* **	5.91 × 10^−6^	
** *SPRR5* **	2.95 × 10^−5^	
** *EFNA3* **	3.38 × 10^−5^	
** *PAX1* **	3.86 × 10^−5^	
** *USP12* **	4.94 × 10^−5^	
**GA + PA vs. Sham + PA**		
**Top Canonical Pathways**	***p*-Value**	**Overlap**
Polyamine Regulation in Colon Cancer	4.95 × 10^−4^	5.1% 3/59
Collagen Degradation	5.45 × 10^−4^	4.9% 3/61
Collagen Chain Trimerization	5.90 × 10^−3^	4.5% 2/44
**Top Diseases**	***p*-Value Range**	**Number of Molecules**
Organismal Injury and Abnormalities	4.98 × 10^−2^–9.96 × 10^−5^	60
Developmental Disorders	4.34 × 10^−2^–3.62 × 10^−4^	15
Connective Tissue Disorders	4.84 × 10^−2^–3.62 × 10^−4^	11
**Molecular and Cellular Functions**	***p*-Value Range**	**Number of Molecules**
Cell Death and Survival	4.98 × 10^−2^–2.61 × 10^−3^	12
Cell-To-Cell Signaling and Interaction	3.63 × 10^−2^–2.61 × 10^−3^	6
Cellular Development	4.59 × 10^−2^–2.61 × 10^−3^	12
Cellular Growth and Proliferation	4.59 × 10^−2^–2.61 × 10^−3^	8
Cellular Movement	4.84 × 10^−2^–2.61 × 10^−3^	8
**Top Networks**		**Score**
Cardiovascular System, Development and Function, Organismal Development, Cardiovascular Disease		53
Cell Death and Survival, Organismal Injury and Abnormalities, Cell Signaling		32
**Top Upstream Regulators**	***p*-Value Range**	
** *HNRNPK* **	2.73 × 10^−4^	
** *HIPK1* **	3.24 × 10^−4^	
N1,N11-diethylnorspermine	7.00 × 10^−4^	
** *TLR5* **	7.22 × 10^−4^	
***Salmonella enterica*** serotype abortus equilibrium polysaccharide	8.09 × 10^−4^	
**GDA + PA vs. Sham + PA**		
**Top Canonical Pathways**	***p*-Value**	**Overlap**
Synaptic Adhesion-like Molecules	3.53 × 10^−4^	14.3% 3/21
TGF- Signaling	4.62 × 10^−4^	5.2% 5/96
Collagen Degradation	7.37 × 10^−4^	6.6% 4/61
**Top Diseases**	***p*-Value Range**	**Number of Molecules**
Cancer	3.93 × 10^−2^–1.08 × 10^−12^	156
Endocrine System Disorders	3.48 × 10^−2^–1.08 × 10^−12^	138
Organismal Injury and Abnormalities	3.93 × 10^−2^–1.08 × 10^−12^	157
**Molecular and Cellular Functions**	***p*-Value Range**	**Number of Molecules**
Cellular Assembly and Organization	3.93 × 10^−2^–2.62 × 10^−4^	29
Gene Expression	3.93 × 10^−2^–3.67 × 10^−4^	41
Cellular Movement	3.93 × 10^−2^–9.06 × 10^−4^	16
Cell Death and Survival	3.93 × 10^−2^–1.37 × 10^−3^	37
Cellular Development	3.93 × 10^−2^–1.37 × 10^−3^	59
**Top Networks**		**Score**
Cardiovascular System, Development and Function, Cellular Development, Cellular Growth and Proliferation		35
Cancer, Organismal Injury and Abnormalities, Developmental Disorders		35
**Top Upstream Regulators**	***p*-Value Range**	
** *SMYD3* **	8.21 × 10^−6^	
** *Firre* **	1.31 × 10^−5^	
** *SPOP* **	1.86 × 10^−5^	
** *FMN2* **	5.11 × 10^−5^	
** *IL2RG* **	5.99 × 10^−5^	
**Placebo + PA vs. Sham + PA**		
**Top Canonical Pathways**	***p*-Value**	**Overlap**
Keratinization	5.89 × 10^−10^	3.3% 7/214
Collagen Degradation	1.40 × 10^−3^	3.3% 2/61
Wound Healing Signaling	1.49 × 10^−3^	1.2% 3/252
**Top Diseases**	***p*-Value Range**	**Number of Molecules**
Gastrointestinal Disease	2.25 × 10^−2^–9.24 × 10^−10^	9
Organismal Injury and Abnormalities	4.89 × 10^−2^–9.24 × 10^−10^	22
Dermatological Diseases and Conditions	4.89 × 10^−2^–3.77 × 10^−7^	10
**Molecular and Cellular Functions**	***p*-Value Range**	**Number of Molecules**
Cell Morphology	1.98 × 10^−2^–5.86 × 10^−6^	3
Cellular Movement	3.79 × 10^−2^–1.03 × 10^−4^	4
Cellular Development	4.63 × 10^−2^–8.46 × 10^−4^	7
Carbohydrate Metabolism	9.10 × 10^−4^–9.10 × 10^−4^	1
Cell Cycle	4.54 × 10^−3^–9.10 × 10^−4^	2
**Top Networks**		**Score**
Gastrointestinal Disease, Organismal Injury and Abnormalities, Dermatological Diseases and Conditions		33
Lipid Metabolism, Molecular Transport, Small Molecule Biochemistry		23
**Top Upstream Regulators**	***p*-Value Range**	
** *POU2F3* **	4.10 × 10^−6^	
** *EFNA4* **	6.35 × 10^−6^	
** *EFNA3* **	7.30 × 10^−6^	
** *PAX1* **	7.80 × 10^−6^	
** *SPRR5* **	1.02 × 10^−5^	

**Table 2 ijms-25-10714-t002:** Top 10 significant differentially expressed transcripts.

Based on False Discovery Rate					
	Upregulated	Downregulated		Upregulated	Downregulated
**Sham + PA vs.** **Sham + Sham**	*SAPCD1*	*TPGS1*	Placebo + PA vs. Sham + PA	*MMP1*	*DNAAF4*
	*BEST1*	*KRT14*		*ZNF837*	*TRPC3*
	*SMOX*	*KRT13*		*CCDC85A*	*TIGD3*
	*AGER*	*FAT4*		*LIPH*	*NUTM2E*
	*TMEM91*	*RAPGEF2*		*SPRR2E*	*GPRIN3*
	*FSCN3*	*POLR2J2*		*PDZRN3*	*PRAM1*
	*RAMP1*	*MIURF*		*NOTCH2NL*	*ST8SIA4*
	*SNX22*	*EEA1*		*HBQ1*	*PADI1*
	*SLC52A1*	*HERC1*		*GRIN3B*	*GARIN1A*
	*ACAP1*	*LPP*		*AL121899.2*	*ITK*
	**Upregulated**	**Downregulated**		**Upregulated**	**Downregulated**
**GA + PA vs.** **Sham + PA**	*MT-ND6*	*FBXL15*	GDA + PA vs. Sham + PA	*MT-ND6*	*FBXL15*
	*FAT4*	*PCSK1N*		*EMP1*	*JUN*
	*RAPGEF2*	*MZT2A*		*FAT4*	*MEX3D*
	*POLR2J2*	*SCAND1*		*GCNT4*	*ATP6C*
	*HERC1*	*TPGS1*		*AHNAK*	*COL6A2*
	*LPP*	*DOHH*		*POLR2J2*	*PCSK1N*
	*POLR2J3*	*CLEC11A*		*HERC1*	*MZT2A*
	*RP11*	*CIMAP1B*		*POLR2J3*	*KLF2*
	*NCR3LG1*	*HES4*		*RP11*	*SBNO2*
	*FAT1*	*PRR7*		*UBR4*	*SCAND1*

## Data Availability

The original data presented in the study are openly available in https://zenodo.org/records/13864634 at DOI: 10.5281/zenodo.13864634 (Accessed on 1 October 2024).

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
