# Peer review of "Global Transcriptomic Analysis of Topical Sodium Alginate Protection against Peptic Damage in an In Vitro Model of Treatment-Resistant Gastroesophageal Reflux Disease"

_ijms, 2024, doi:10.3390/ijms251910714_

Round 1

Reviewer 1 Report (Previous Reviewer 1)

Comments and Suggestions for Authors

No further comments.

Author Response

Dear reviewer,

Thank you for your valuable reviews.

Best regards.

Reviewer 2 Report (Previous Reviewer 2)

Comments and Suggestions for Authors

I appreciate the effort of the Authors to revise the manuscript. Nevertheless, according to the current global standards, the raw data of high-throughput evaluation (omics) should be deposited (raw files) in appropriate repositories (e.g. GEO2R) very often with specific separate DOI or accession number. Supplementary data (as stated by Authors) contains supplementary analysis but does not contain the raw file.

Nevertheless, I leave it to the decision of the Editor whether IJMS requires to follow this guideline.

Author Response

Dear reviewer,

Thank you for your valuable reviews. We have uploaded our raw data to Zenodo.org (https://zenodo.org/records/13864634, DOI: 10.5281/zenodo.13864634). A statement has been added to the manuscript in blue color.

Best regards.

This manuscript is a resubmission of an earlier submission. The following is a list of the peer review reports and author responses from that submission.

Round 1

Reviewer 1 Report

Comments and Suggestions for Authors

I don't have any major comments for the manuscript. It can be published after editorial proofreading. One thing to mention is that the manuscript could be improved by discussing a potential role of the microbiota in Gastroesophageal Reflux Disease as the model system the authors used for RNA-seq in the manuscript was an in vitro model which does not fully recapitulate humans. The discussion could also be improved if the authors further discuss possible implications of their in vitro findings in humans.

Author Response

Comment 1:

I don't have any major comments for the manuscript. It can be published after editorial proofreading. One thing to mention is that the manuscript could be improved by discussing a potential role of the microbiota in Gastroesophageal Reflux Disease as the model system the authors used for RNA-seq in the manuscript was an in vitro model which does not fully recapitulate humans. The discussion could also be improved if the authors further discuss possible implications of their in vitro findings in humans.

Answer 1:

Dear Reviewer,

We appreciate your valuable suggestions and have incorporated a discussion on the potential role of the microbiota, specifically Helicobacter pylori (H. pylori), in GERD, as you recommended. We recognize that our in vitro model does not fully capture the complexity of human disease, particularly the interactions between H. pylori and GERD. As you mentioned, H. pylori can exacerbate GERD symptoms and influence mucosal integrity and inflammation. There is also emerging evidence suggesting that sodium alginate may have a beneficial effect on H. pylori, potentially reducing its impact on GERD progression. We have expanded the limitations section to address the exclusion of H. pylori from our model, acknowledging that this limits our understanding of how alginate formulations might interact with both GERD and H. pylori-related pathology.

Furthermore, we have added a discussion to the limitation paragraph on the implications of our in vitro findings in human populations, particularly highlighting how pepsin-induced dysregulation of oncogenic, inflammatory, and ECM-related pathways may contribute to GERD progression, especially in patients with persistent weak acid reflux despite acid suppression. The observed protective effects of Gaviscon formulations in vitro may translate into clinically relevant benefits, potentially reducing epithelial damage and lowering the risk of progression to more severe conditions such as Barrett’s esophagus or esophageal adenocarcinoma. We agree that further clinical studies are necessary to validate these findings in human populations.

Thank you again for your insightful feedback, which has helped us improve the clarity and depth of the manuscript.

Reviewer 2 Report

Comments and Suggestions for Authors

This study provides high throughput data showing transcriptomic changes in esophageal cell line Het-1A exposed to various types of treatments. I see this as an important aspects valuable to the development of esophageal pathophysiology-oriented research. There are many limitations which are however fairly addressed in appropriate section of the manuscript. Based on our broad experience covering in vitro models of esophageal pathologies, I would recommend other cell lines that were shown by recent studies as more appropriate to mimick clinical course of upper GI tract pathologies, e.g. EPC2 or NES-G2T. AS an asset, I consider the implementation of the treatment with GDA.

As a limitation, I would add also the information that reflecting this study is only “descriptive analysis” and there is no functional confirmation of the transcriptomic analysis outcomes.

Perhaps, I have overlooked this in the text but it is required to upload the sequencing raw data to appropriate database, to make it publicly available and to implement appropriate links/numbers in the main text of the manuscript.

Author Response

Comment 1:

This study provides high throughput data showing transcriptomic changes in esophageal cell line Het-1A exposed to various types of treatments. I see this as an important aspect valuable to the development of esophageal pathophysiology-oriented research. There are many limitations which are however fairly addressed in appropriate section of the manuscript. Based on our broad experience covering in vitro models of esophageal pathologies, I would recommend other cell lines that were shown by recent studies as more appropriate to mimic clinical course of upper GI tract pathologies, e.g. EPC2 or NES-G2T. AS an asset, I consider the implementation of the treatment with GDA.

As a limitation, I would also add the information that reflecting this study is only “descriptive analysis” and there is no functional confirmation of the transcriptomic analysis outcomes.

Perhaps, I have overlooked this in the text, but it is required to upload the sequencing raw data to appropriate database, to make it publicly available and to implement appropriate links/numbers in the main text of the manuscript.

Response 1:

Dear Reviewer 2,

Thank you for your thorough review and insightful suggestions. We appreciate your positive feedback on the transcriptomic data and the recognition of its value in esophageal pathophysiology research.

Regarding the use of alternative cell lines, such as EPC2 or NES-G2T, we acknowledge that these models may more closely mimic the clinical course of upper GI tract pathologies, as suggested by recent studies. While we utilized the Het-1A cell line for its established relevance in GERD research, we agree that exploring these other cell lines in future studies could provide additional insights and validate our findings further.

In response to your comment about the "descriptive analysis" and the lack of functional confirmation of the transcriptomic results, we have updated the limitations section to clearly state that our study is primarily descriptive. We recognize that functional assays are necessary to confirm the biological significance of the observed transcriptomic changes, and this will be a focus in our subsequent research.

Regarding the raw sequencing data, we have included the raw data in the supplementary material of the manuscript. Although the data is not publicly accessible via an external database, we have provided the data in an accessible format within the supplementary materials section, as stated in the manuscript.

Thank you again for your valuable feedback, which has significantly contributed to improving the manuscript and guiding our future research directions.